# A Proposed System for Multi-UAVs in Remote Sensing Operations

**DOI:** 10.3390/s22239180

**Published:** 2022-11-25

**Authors:** Pablo Flores Peña, Marco Andrés Luna, Mohammad Sadeq Ale Isaac, Ahmed Refaat Ragab, Khaled Elmenshawy, David Martín Gómez, Pascual Campoy, Martin Molina

**Affiliations:** 1Department of System Engineering and Automation, University Carlos III of Madrid, 28911 Leganés, Spain; 2Drone-Hopper Company, 28919 Leganés, Spain; 3Computer Vision and Aerial Robotics Group, Centre for Automation and Robotics (CAR), Universidad Politécnica de Madrid (UPM-CSIC), 28006 Madrid, Spain; 4Wake Engineering Company, 28906 Getafe, Spain; 5Department of Electrical Engineering, University Carlos III of Madrid, 28919 Leganés, Spain; 6Department of Network, Faculty of Information Systems and Computer Science, October 6 University, Giza 12511, Egypt; 7Department of Artificial Intelligence, Universidad Politécnica de Madrid, 28040 Madrid, Spain

**Keywords:** UAV-Swarm, coverage path planning, remote sensing, control strategies

## Abstract

This paper proposes the design of the communications, control systems, and navigation algorithms of a multi-UAV system focused on remote sensing operations. A new controller based on a compensator and a nominal controller is designed to dynamically regulate the UAVs’ attitude. The navigation system addresses the multi-region coverage trajectory planning task using a new approach to solve the TSP-CPP problem. The navigation algorithms were tested theoretically, and the combination of the proposed navigation techniques and control strategy was simulated through the Matlab SimScape platform to optimize the controller’s parameters over several iterations. The results reveal the robustness of the controller and optimal performance of the route planner.

## 1. Introduction

In the last decade, Unmanned Aerial Vehicles (UAVs) have gained a lot of attention from researchers due to their huge benefits. The concept of unmanned aerial vehicles reflects the fact that it is a plane without a pilot on board, meaning that all the interaction with and control of this vehicle takes place remotely; in addition, the term has recently been changed to become Unmanned Aerial Systems (UAS) to reflect the complexity of the interactions between the UAV, the Ground Control Station (GCS), and the telecommunication system in operating a UAV. The field of UAVs covers many aspects, including military and civilian applications, where military applications can be used in reconnaissance, attack roles, rescue missions, and training of military troops, while civilian applications can be used in industrial support [1], fire extinguishing [2], logistics delivery, and autonomous transport.

Different high-level technologies are used throughout UAV missions, starting from communication, networking, controlling, and aerodynamics; all these technologies are combined to form a unique platform capable of meeting the mission’s specific requirements. In addition, each UAV can be equipped with a special payload that depends upon several factors, such as the real payload availability for the UAV (how much it can carry), and the mission’s requirements, such as thermal cameras, HD cameras, special sensors, etc. In addition, each mission depends upon the UAV’s endurance, range, altitude, speed, and payload.

In addition, the benefits of using UAVs in different missions has led researchers to investigate how to overcome the missions’ problems by dealing with different techniques such as the swarming technique [3], where the word SWARM comes from nature, such as how bees fly and attack in swarms. Thus, through swarm missions, the type of mission and the payload can be split between the UAVs, so that what was once achieved over multiple missions can now be performed in a single mission using the swarming technique.

Through this research, a new swarming technique algorithm will be discussed in detail, showing the advantages of using such algorithms. The swarming technique will be simplified into four main layers: the communication layer, the networking layer, the decision-making layer, and finally, the path-planning layer.

Furthermore, through the communication layer and the network layer, UAV swarms are concurrently controlled by the ground control station (GCS). Complex software is run through the GCS, and it is equipped with a communication system (transceiver) that propagates the telemetry data (sending and receiving) through all connected UAVs to the GCS, achieving system requirements. Moreover, the telemetry data include several parameters such as ground speed, GPS information, gyroscopes, and payload data coming from cameras and sensors (accelerometers). Commonly, the communication system uses unlicensed radio frequency bands such as 433 Mhz/1 W(+30 dBi)/−116 dBm or 866–950 Mhz/1000 mW(+30 dBm)/−116 dBm for sending and receiving data.

In addition, adding an onboard computer to the UAVs, such as a Jetson Xavier [4], will provide the system with a higher level of autonomy, allowing the system to act as a decision-making system so that each UAV will act as a PC by itself.

In [4], the authors stated that a swarm communication architecture could be classified into two main architectures: an ad hoc network-based architecture and an infrastructure-based swarm architecture, as shown in Figure 1 and Figure 2.

The ad hoc networks can be classified into four main categories: mobile ad hoc networks (MANET), vehicle ad hoc networks (VANET), flying ad hoc networks (FANET), and underwater ad hoc networks (UWVANET), as stated in [3]. Correspondingly, UAVs are a sub-category in the ad hoc networks, as they are a category within FANETs with the following characteristics: the node mobility is very high; the mobility model is regular for predetermined paths but special mobility models exist for autonomous multi-UAV systems; the node density is very low; the topology change is fast; the radio propagation model is high above the ground, where the Line of Sight (LOS) is available for most cases; the computational power is high; and finally, the localization is achieved through GPS, AGPS, DGPS, or IMU.

Additionally, an integral part of any swarm application is the control strategy to maintain the stability of each UAV and the system as a whole for a homogeneous flight. To this end, the authors of [5] presented a robust controller for a tracking mission lying on a decentralized multi-agent platform; they chose two potential functions, one quadratic and another with attraction and repulsion parts, to be compared when exposed to the controller. Their results affirm that the sliding mode controller behaves fantastically, independent of the potential functions and the tracking term aims for the controller to follow the target. In [6], the authors performed a numerical study of robust algorithms for a time-varying swarm with nonlinear external errors. To solve the instabilities, they first expanded the observation matrix for a consensus problem, and the “n” employing a robust compensator and a nominal controller, the system converged to stability with small errors; however, their solution does not support a dynamic switch formation, and no practical work was performed. In [7], however, a robust controller was designed to regulate the behavior of two swarm groups under conflict, formulated as persuaders following evaders: first, one individual of each group follows another individual, and second, the whole group follows another group.

Meanwhile, the drag force triggered by the evaders and their turning effects is considered an uncertainty. Their study reveals that the robust method hastens system convergence, which is advanced by increasing the controller gain parameter; however, the chattering phenomenon is observed in the turning modes. Considering previous studies, the current paper presents a novel robust algorithm that reduces chattering by providing a smooth effect on inconsistent controller outputs.

Moreover, for the decision-making layer, multiple types of research in the state-of-art deal with the task-assignment problem [8,9] and mission planning [10,11] through UAV swarm systems. In this context, multi-UAV coverage path planning (CPP) is a line of research whose objective is to find the optimal route to cover a defined area and the optimal path allocation for each UAV [11]. This field has gained the attention of researchers due to its multiple remote sensing applications such as in precision agriculture [12,13], inspection [14,15], search and rescue [16,17], and others. In general terms, the research aims to find an optimal path given the set of points of a polygonal area. However, other applications could require covering multiple disjoint zones during the same operation. On the other hand, the approach presented in [18] proposes an asymptotically optimal coverage navigation algorithm for a multi-UAV system for non-polygonal regions; however, the authors do not deal with the problem of separated zones.

For this reason, researchers in [19] have introduced a new concept, the Traveling Salesman Problem Coverage Path Planning (TSP-CPP), whose goal is to optimize the routes and the order of visits to the target areas. Some strategies for single UAV systems have been presented in the literature [19], where the researchers use dynamic programming and a grid-based approach. In [20], the researchers apply a back-and-forth optimization strategy to find the routes and genetic algorithms (GA) to search the order of visits. In addition, for multi-UAV systems, the technique presented in [21] can be mentioned, which solves the problem using evolutionary methods; as an NP-hard problem, these strategies can consume a lot of computational resources depending on their dimensionality. This paper presents a complete algorithm for solving the TSP-CPP problem in multiple areas using fast-optimization techniques, representing a new planning system that is fast enough to execute in-flight replanning.

This paper is organized as follows: Section 2 represents the materials and methods used, where the state-of-art is shown clearly; Section 3 represents the results obtained; Section 4 discusses the results obtained in Section 3; and finally, Section 5 concludes the paper.

## 2. Materials and Methods

### 2.1. Control Strategy

Considering 6 degrees of freedom (DoF) for the dynamic system, a novel robust controller is presented, consisting of a robust compensator and a nominal controller. Meanwhile, the system is assumed to be an output time-varying model such as $H\left(t\right)\in R^{n}$ due to unknown uncertainties, namely, changing swarm trajectory dynamically, which also necessitates the controller to include a decision-making part. The dynamic system definition of each agent i∈{1,2⋯,n} in the corresponding swarm group is given by:(1)$$\left\{\begin{array}{c} \dot{X}\left(t\right)=AX_{i}\left(t\right)+B\left(U_{i}\left(t\right)+N_{i}\left(t\right)\right)\\ Y_{i}\left(t\right)=CX_{i}\left(t\right) \end{array}\right.$$
where $X\left(t\right)\in R^{n}$ is the state vector of the system, including n controller variables; $A\in R^{n\times n}$ is the Lagrangian system matrix; $B\in R^{n\times m}$ is a full rank matrix of m column input parameters; $U\in R^{m}$ is a full rank vector of m row input parameters; $N\in R^{m}$ is a full rank of m row uncertainty parameters; $C\in R^{m}$ is a full rank vector of m row output parameters; and *Y* is the output. Then, rewriting the controller input as $U_{i}\left(t\right)=U_{i}^{nom}\left(t\right)+U_{i}^{rob}\left(t\right)$ standing for nominal and robust parts. Solving for the Laplacian transformation for a time-varying output model, as mentioned in [6,7], gives:(2)$$\left\{\begin{array}{c} U_{i}^{nom}\left(t\right)=\lambda_{1}Y_{i}\left(t\right)+\lambda_{2}\left(Y_{i}\left(t\right)-H_{i}\left(t\right)\right)+\lambda_{3}U_{i}\left(Y_{i}\left(t\right)-H_{i}\left(t\right)\right)\\ U_{i}^{rob}\left(t\right)=-L^{-1}\left({\left(1-\Omega \left(s\right)\right)}^{-1}\Omega \left(s\right)\left(N_{0}^{-1}\left(s\right)D_{0}\left(s\right)LY_{i}\left(t\right)-LU_{i}^{nom}\left(t\right)\right)\right) \end{array}\right.$$
where *L* and $L^{-1}$ are the Laplacian transformation and transform inverse, respectively; the term $N_{0}^{-1}\left(s\right)D_{0}\left(s\right)$ is a standard remarking for Laplacian transformation matrix; and Ω(s) is the Laplacian form of ${\left(\frac{\omega }{s+\omega }\right)}^{a}$, in which both ω and a are positive constants equal to or bigger than the transformation matrix power. Following the solution performed in previous works [22,23,24], controller update parameters that update the system’s error ($e_{i}$) matrix could be defined as $\theta_{i}$. Therefore, the state space formulas can be written as follows:(3)$$\left\{\begin{array}{c} {\dot{X}}_{e}\left(t\right)=AX_{e}\left(t\right)+B_{e}\left(1-\Omega \left(t\right)\right)\\ E\left(t\right)=Y\left(t\right)=CX_{e}\left(t\right) \end{array}\right.$$

Furthermore, the sliding surface equation can be obtained as $S\left(t\right)=\dot{E}+\Omega E$. Updating the error matrix and considering the state matrix as planar and vertical movements and rotations in the form of Euler angles, the control logic could be stated as given:(4)$$\left\{\begin{array}{c} {\ddot{X}}_{e}\left(t\right)=-\Omega {\dot{X}}_{e}\left(t\right)-K\\ E\left(t\right)=Y\left(t\right)=CX_{e}\left(t\right) \end{array}\right.$$

### 2.2. Multi-UAV Mission Management

In this study, the mission management system involves the decision-making and path-planning systems. The ground control station (GCS) calculates the best routes and assigns the task for each UAV in the proposed architecture. The objective of this subsystem is to receive high-level commands from the operator and translate them into a set of waypoints for each UAV, optimizing the mission execution time. Thus, this system depends on the communication layer to transmit the commands to the UAVs and receive their state for making decisions; it also depends on the control layer to ensure flight stability during the mission.

#### 2.2.1. Problem Statement

Given a set of **K** UAVs and a set of polygonal areas **A** defined in the $R^{2}$ space, the idea is to find the individual routes for each UAV to cover all the areas in the shortest time as shown in Figure 3.

For this problem, it is assumed that the number of polygons to be covered is different from the number of UAVs; moreover, the areas of the polygons are different. It is assumed the back-and-forth (BF) pattern [25] is the coverage route to simplify the path design.

Therefore, according to the problem features, the main task can be divided into subtasks such as defining the order of area visits, calculating the shortest route that covers the set of polygonal areas, and distributing the mission among the UAVs. In addition, all UAVs take off at the same time from different locations. In remote-sensing applications, an important parameter to consider is the payload footprint. In multi-UAV systems, one strategy to ensure proper operation is that the UAVs are homogeneous and have the same payload; in addition, they should all fly at the same altitude, as shown in Figure 4.

Thus, the method to calculate the spacing between waypoints and spacing between lanes will depend on the payload footprint, but its calculation is not the aim of this study.

#### 2.2.2. Coverage Path Planning Algorithm

The algorithm used in this study consists of four stages, such as computing the order of visiting each area, calculating the BF path, and assigning optimal UAV routes. This strategy combines the concepts proposed by [20,26] to develop a new approach for a multi-UAV system.

Thus, to determine the order in which each area is visited, a solution is proposed based on a simple TSP solution by using the centroids of each area as a city to visit and the centroid of UAV positions as deployment locations. The process is shown in Figure 5.

Figure 5a shows the areas and the UAV positions, each area’s centroids, and the centroid of UAV locations (Figure 5b). The solution of the TSP is found using the simulated annealing algorithm [27], as shown in Figure 5c.

The calculation of the BF path is performed using distance optimization to find the optimal sweep line direction using a Powell optimization algorithm [28] given by Equation (Equation 5):(5)$$\sum_{i=0}^{n}BF\left(A_{1},\theta_{1}\right)+BF\left(A_{2},\theta_{2}\right)+\cdots +BF\left(A_{n},\theta_{n}\right)$$
where BF is the back-and-forth function that depends on area A and the sweep line direction θ. Then, the areas are joined, taking into consideration the last point of the back and forth of the previous area and the first of the next area, as shown in Figure 6.

Finally, the multi-UAV task assignment is conducted by using the Powell-BINPAT algorithm described in our previous research [26], as shown in Figure 7.

This method takes the entire path and distributes it optimally among the UAVs in the swarm. Additionally, in contrast to the other techniques presented in the literature, BINPAT plans take into account different take-off locations for each UAV and the intermediate points of the path; the planning based on these criteria is useful when analyzing the footprint of a remote sensing device.

## 3. Results

### 3.1. Numerical Results

To test the algorithms for area coverage applications, two to four UAVs were flown in two to four areas, as shown in Figure 8 regarding graphical results.

In Figure 7, the algorithm’s performance is not affected by the size of the regions; the Powell-BINPAT algorithm is robust to area size changes. When there are fewer regions than UAVs, this technique will determine which zones need to be covered by more UAVs; otherwise, the algorithm determines which UAVs can cover more areas. For analysis purposes, the cost of a mission has been measured in terms of distance traveled, and the coefficient of variation is calculated according to Equation (Equation 6):(6)$$CV=\frac{SD}{Av}*100$$
where SD is the standard deviation, and Av is the average distance. The results are presented in Table 1.

In Table 1, the results show that the coefficient of variation tends to increase when there are more regions to explore, and it also increments proportionally to the number of UAVs. When the number of UAVs increases, the distances to be covered by each aircraft are smaller, so the standard deviation grows.

### 3.2. Simulation Results

A complete simulation is performed assuming a mission scenario, as shown in Figure 9. The start and end points are the same at (0,0), and three drones are chosen to manage the swarm mission. In addition, the mission zones are drawn randomly in polygons, which, in this study, are four separate zones. Meanwhile, as shown in Figure 9, the colored routes are those generated by the algorithm, in which each color belongs to an individual drone.

Advancing the swarm application, the MATLAB Simulink platform powered the simulation process to examine the controller’s functionality during the flight. Approaching the practical tests, some assumptions are considered during the simulation: the system solver is discrete with a fixed-step type of 0.001 s, the average cruise velocity is 5 m/s, and the flight altitude is up to 20 m, which needs a longer time in comparison to the variable-step mode with an ordinary differential equation (ODE) solver, but the results are much more accurate to the reality. To this end, Figure 10 shows the overall diagram of a swarm system consisting of three drones flying in distinct zones but with similar relative distances as practical tests. Once the decision-making system defines the waypoints, the reference values of the controller are desired, which are the position and heading angles. Secondly, the attitude desired values are generated, and finally, the motor mixing part interprets the commands for the motors (see [22,23,24]).

As shown in Figure 10, the robust controller receives the state feedback to reduce the actual errors. The drone boxes are the 3D CAD components imported to the virtual SimScape space, considering contact forces, revolt joint frictions, and a 6DoF model applied to the drones’ models. Explicitly, the control’s lower-level loop is shown in Figure 10, and as two modes emphasizes, the desired values are generated first through the reference generator, and then the robust parameters are multiplied by the errors of the value itself and its derivatives that finally are exported as commanded moments.

As shown in Figure 11, the function is to smooth the chattering zone using the stability approach. Afterward, the motor mixing box saturates the outbound inputs of the controller and distributes them into four motors as defined for every quadcopter, and the dynamic multi-body receives the command to interpret them as SimScape logical forces and moments. Following the swarm control algorithm, simulation results are shown in Figure 12.

As shown in Figure 12, the drones follow the reference trajectories when the swarm decision-making system commands, which could even be dynamically changed during the mission. To this end, the controller algorithm is designed to face changing parameters, namely, the weight and the trajectory of the swarm. In addition, the time and distance data of the simulation are elaborated in Table 2. Comparing the results, the decision-making algorithm is functionally optimized so that the total time difference is less than 9 s among the UAVs, and the total trajectory distances are managed to obtain similar values.

## 4. Discussion

The results presented in the previous section demonstrate the graphically robust behavior of the sliding mode controller when facing various clients and several objectives, which are stated as distinct drones receiving different waypoints. In this context, by generating the waypoints using the decision-making system as the high guidance level and then controlling the drones’ attitude system as the low level, the swarm application is optimized, as it is also possible to check the optimization of the trajectory assignment. The low CV values calculated and reported in Table 1 show the optimal behavior of the path planning algorithm because the mission is performed in the shortest possible time. In addition, the energy expenditure of each UAV is almost homogeneous; this can be corroborated in Table 2, where the simulation results report similar mission times and traveled distances for the case of three UAVs and four explored areas.

## 5. Conclusions and Future Work

The presented research deals theoretically and in simulation with the main problems of a multi-UAV system in remote sensing operations with new proposals. The results of the TSP-CPP techniques show an optimal task allocation for each UAV based on the mission cost, and the control system shows robustness when maneuvering in the proposed route. With these theoretical results, the study of the sensors to be chosen as payload depending on the task to be performed by the UAV can be improved. In future work, we propose the real implementation of this system and the analysis of the sensors in the payload.

## Figures and Tables

**Figure 1 sensors-22-09180-f001:**
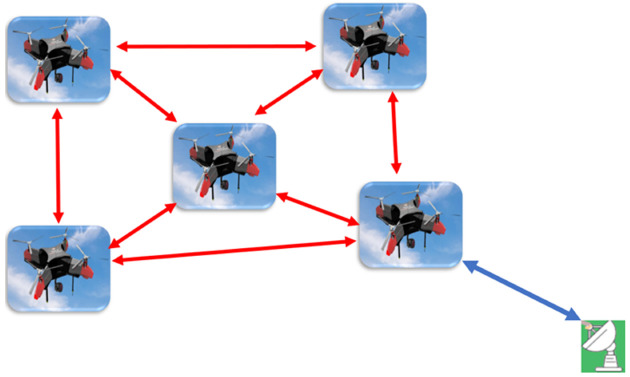
Communication architecture of UAV swarms based on FANET.

**Figure 2 sensors-22-09180-f002:**
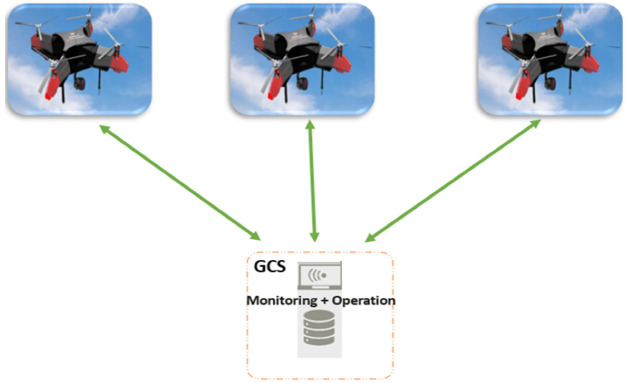
Block diagram of infrastructure (GCS)-based swarm architecture.

**Figure 3 sensors-22-09180-f003:**
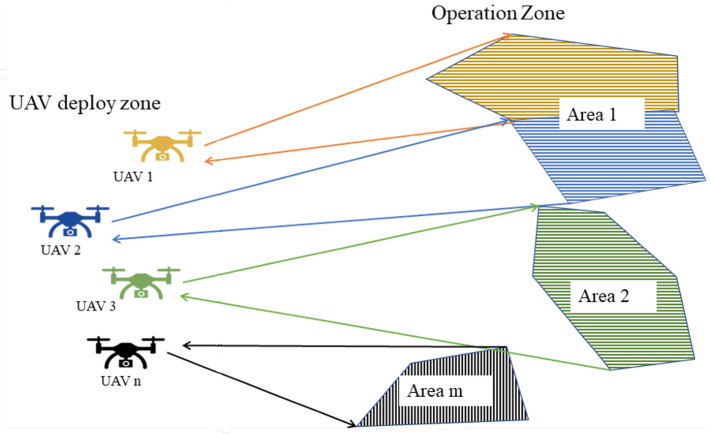
Mission operation schema.

**Figure 4 sensors-22-09180-f004:**
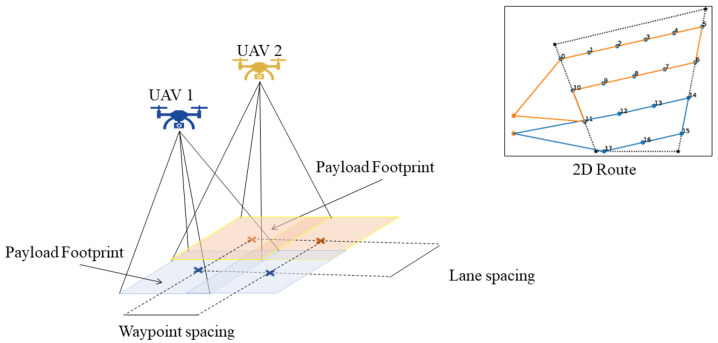
Footprint schema in 2D routing mission.

**Figure 5 sensors-22-09180-f005:**
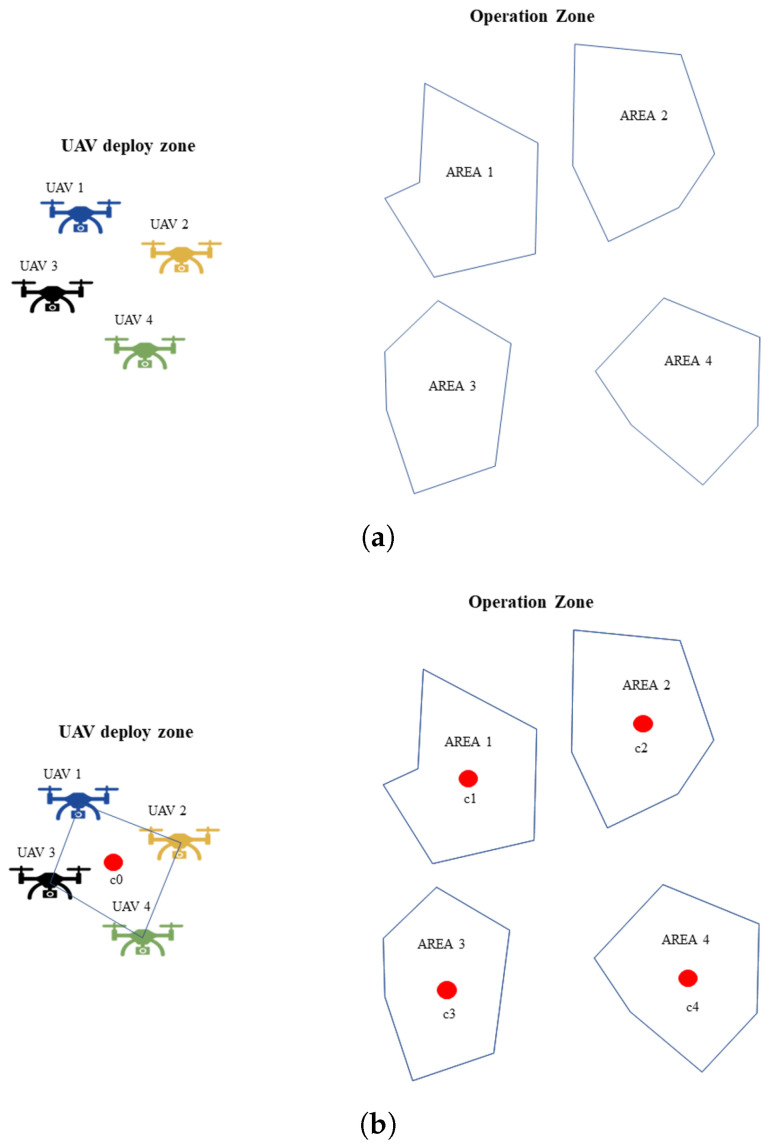
A schematic of the TSP applied to the multi-UAV system. (**a**) Initial problem statement. (**b**) Centroids calculation in UAV locations and multiple regions. (**c**) TSP solution found in centroids.

**Figure 6 sensors-22-09180-f006:**
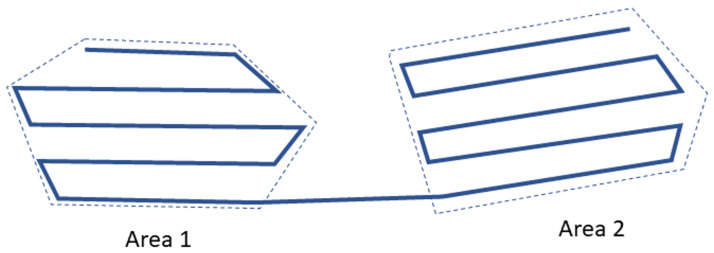
Schema of area joining criteria.

**Figure 7 sensors-22-09180-f007:**
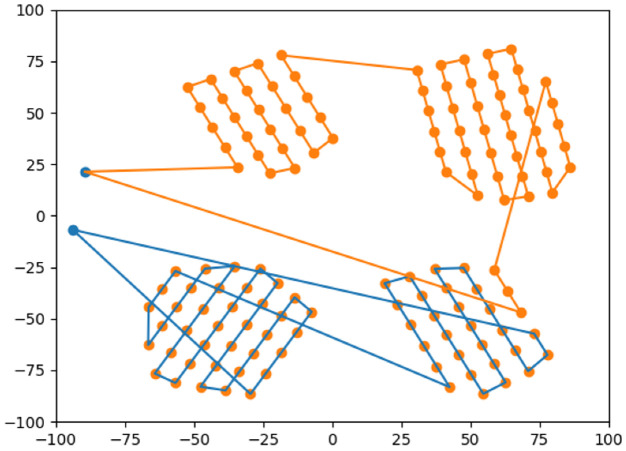
Powell-BINPAT task assignment.

**Figure 8 sensors-22-09180-f008:**
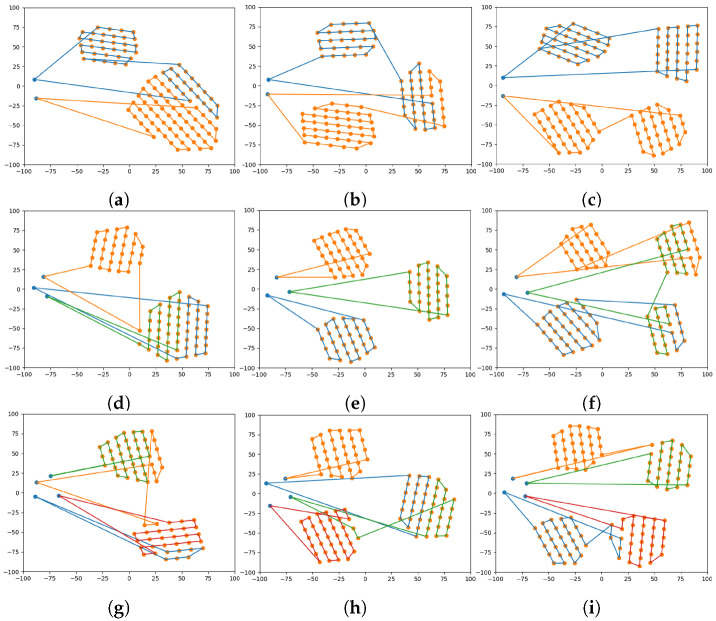
Results of the path-planning system for multiple UAVs and multiple areas: (**a**) two UAVs and two areas; (**b**) two UAVs and three areas; (**c**) two UAVs and four areas; (**d**) three UAVs and two areas; (**e**) three UAVs and three areas; (**f**) three UAVs and four areas; (**g**) four UAVs and two areas; (**h**) four UAVs and three areas; (**i**) four UAVs and four areas.

**Figure 9 sensors-22-09180-f009:**
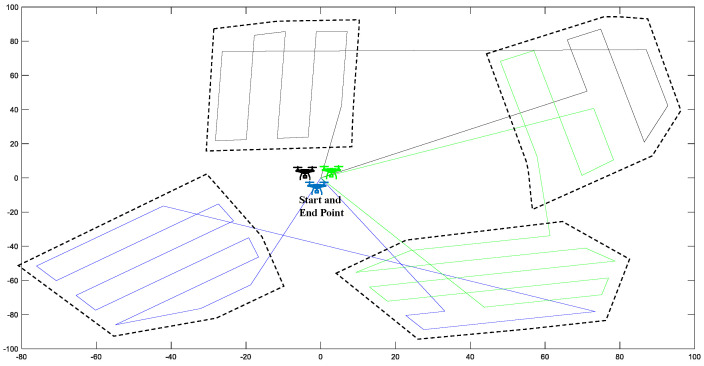
Powell-BINPAT task assignment.

**Figure 10 sensors-22-09180-f010:**
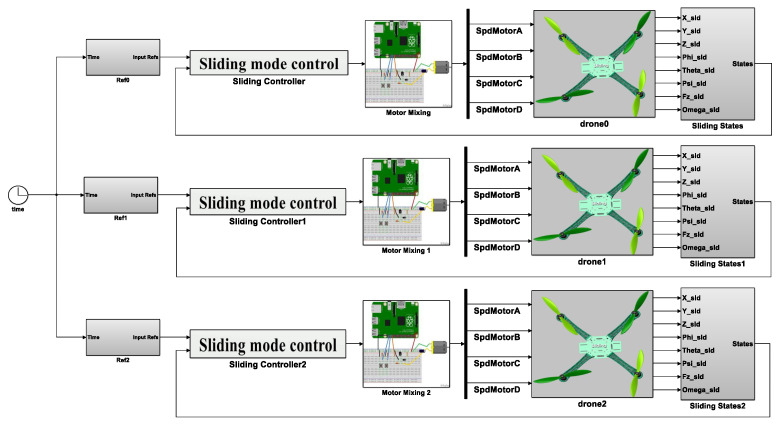
The simulation diagram of three drones in a swarm application.

**Figure 11 sensors-22-09180-f011:**
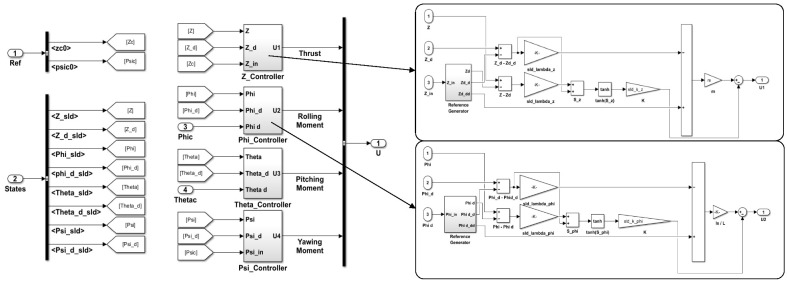
The simulation diagram of three drones in a swarm application.

**Figure 12 sensors-22-09180-f012:**
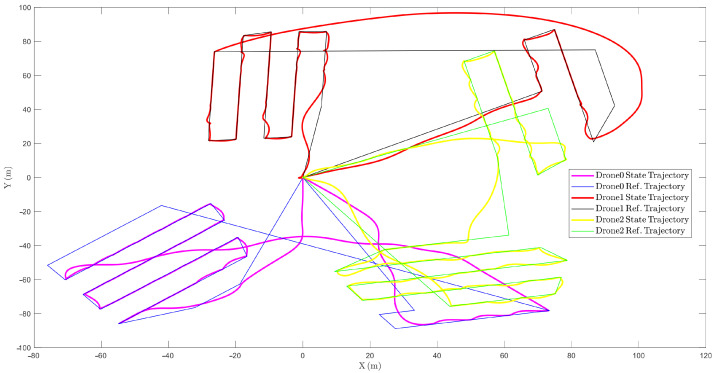
The simulation results of three drones in a swarm application.

**Table 1 sensors-22-09180-t001:** Distance results for multiple UAVs and multiple areas to explore.

No. UAVs	No. Areas	UAV 1	UAV 2	UAV 3	UAV 4	Av	SD	CV
2	2	702.58	707.39	-	-	704.98	3.40	0.48
	3	925.25	908.20	-	-	916.73	12.05	1.32
	4	886.39	952.45	-	-	919.42	46.70	5.08
3	2	557.71	496.80	529.41	-	527.97	30.48	5.77
	3	626.84	569.27	659.26	-	618.46	45.58	7.37
	4	792.35	675.95	782.72	-	750.34	64.60	8.61
4	2	472.83	395.66	416.31	490.24	443.76	44.99	10.14
	3	430.83	494.15	489.77	513.73	482.12	35.74	7.41
	4	586.58	546.09	509.70	592.72	558.77	38.71	6.93

**Table 2 sensors-22-09180-t002:** Numerical results of the swarm simulation.

No. UAVs	UAV id	Total Time (s)	Total Distance (m)
3	UAV1	178.106	888.954
No. Areas	UAV2	169.203	845.994
4	UAV3	173.627	867.694

## Data Availability

Regarding the numerical analysis performed for this paper a simulation video is prepared: https://drive.google.com/file/d/1oepowt6ssy59VIh9bJnO-A_fgvNnXzSQ/view?usp=sharing.

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
