# Peer review of "A Proposed System for Multi-UAVs in Remote Sensing Operations"

_sensors, 2022, doi:10.3390/s22239180_

Round 1
Reviewer 1 Report
This is an interesting research about the design of communications, control systems, and navigation algorithms of a multi-UAV system focused on remote sensing operations. A new controller based on a compensator and a nominal controller is designed to regulate the UAVs’ attitude dynamically. If this method can be can be widespread, the surveying efficiency by using the UAV will be enhanced.
However, before it can be published on this journal, some part of it should be polished.
(1) All the figures are fuzzy, all of them should be replaced.
(2) Discussion should add more information, especially the factors that influenced the efficiency and the precision should be analyzed.
(3) Video about the operation should added in the supplement material. Because it speaks louder than the words in this manuscript.

Reviewer 2 Report
In this paper, a new controller based on compensator and nominal controller is designed to dynamically adjust the attitude of unmanned aerial vehicle (UAV), so as to reduce the chatter in turning mode and achieve robust effect. At the same time, the navigation system addresses the multi-region coverage trajectory planning task using a new approach to solve the TSP-CPP problem. In addition, the author simulated the combination of the navigation technology and control strategy on the Matlab SimScape platform. The results show the robustness of the controller and optimal performance of the route planner.
Generally speaking, the research method of this paper is clear and structured. But I have a few trivial suggestions for your reference. I hope the author can improve the following parts.
1. In the article, the order of the two group communication architectures in line 51 is opposite to the figure order. And the figure 1 is a little fuzzy.
2. The introduction can be more organized and clearer, and the non-key points of the article can only be briefly introduced.

Round 2
Reviewer 1 Report
The authors have revised the manuscript based on my suggestions.